# Water Management for Construction: Evidence for Risk Characterization in Community and Healthcare Settings: A Systematic Review

**DOI:** 10.3390/ijerph17062168

**Published:** 2020-03-24

**Authors:** Molly M. Scanlon, James L. Gordon, William F. McCoy, Melissa F. Cain

**Affiliations:** 1Phigenics, LLC, 3S701 West Avenue, Suite 100, Warrenville, IL 60555, USA; wmccoy@phigenics.com (W.F.M.); mcain@phigenics.com (M.F.C.); 2Department of Community, Environment, and Policy, Mel and Enid Zuckerman College of Public Health, University of Arizona, Tucson, AZ 85724, USA; 3Gordon Architectural Design, Coronado, CA 92118, USA; jimgordonarchitect@yahoo.com

**Keywords:** water management, water safety, construction, *Legionella*, nontuberculous mycobacteria (NTM), risk characterization, waterborne pathogens, prevention

## Abstract

Construction activities are a known risk contributing to the growth and spread of waterborne pathogens in building water systems. The purpose of the study is to integrate evidence for categorizing construction activity risk factors contributing to waterborne disease in community and healthcare settings, establish severity of such risk factors and identify knowledge gaps. Using a systematic review, the inclusion criteria were: (1) studies with disease cases suspected to be associated with construction activities and waterborne pathogens, and (2) active construction work described in a community or healthcare setting. Each construction activity risk factor was correlated across all studies with the number of disease cases and deaths to establish risk severity. The eligibility review and quantitative synthesis yielded 31 studies for inclusion (community, n = 7 and healthcare, n = 24). From 1965 to 2016, a total of 894 disease cases inclusive of 112 deaths were associated with nine construction activity risk factors and waterborne pathogens. The present study findings support the need for building owners, water management teams and public health professionals to address construction activity risk factors and the analysis of current knowledge deficiencies within the scope of an ongoing water management program. The impact of construction activities on waterborne disease is preventable and should no longer be considered incidental nor accidental.

## 1. Introduction

Construction activities are a known risk contributing to the growth and spread of waterborne pathogens in building water systems [1,2,3]. From 2000–2014 during outbreak investigations for Legionnaires’ Disease (LD) the Centers for Disease Control and Prevention (CDC) found environmental deficiencies (process failures, 65%; human error, 52%; equipment failures, 35% and unmanaged external changes, 35%) negatively impacted building water systems increasing illness, injury and death [4]. The category entitled unmanaged external changes included risk from construction which in-turn increased harm from physical, chemical and microbial water hazards. The CDC defined an unmanaged external change as outside of the building owner’s control which impacted the building water system; hazard controls were not considered. The unmanaged external change category identified two primary environmental conditions—natural disasters (25%) and construction activities (75%). Weather conditions from natural disasters (e.g., tropical storms or flooding) are often unpredictable. However, construction activities are known events with planned activities occurring within specific time frames. The principal investigators of this study found no systematic review summarizing the state of the scientific evidence on water safety before, during or after construction activities. For building owners with projects under construction, the specifics related to the construction activity risk factors, severity of risk, populations at risk, waterborne pathogens of interest, number of associated disease cases and deaths and environmental conditions contributing to the problem remain largely unknown.

Additionally, since 2017, inpatient healthcare settings (acute care hospitals, critical access hospitals and long-term care facilities) seeking Centers for Medicare and Medicaid Services (CMS) reimbursement are obligated to comply with the CMS Survey and Certification (S&C) Group Memorandum to establish a water management program (WMP) [5]. The CMS S&C 17–30 Memo Requirement to Reduce *Legionella* Risk in Healthcare Facility Water Systems to Prevent Cases and Outbreaks of LD is the first USA Federal policy directing healthcare providers to comply with implementation of a WMP [6]. The CMS S&C 17–30 Memo identifies *Legionella* and other opportunistic waterborne pathogens (e.g., *Pseudomonas*, *Acinetobacter*, *Burkholderia*, *Stenotrophomonas*, nontuberculous mycobacteria (NTM) and fungi—*Fusarium* and *Aspergillus*) as pathogens of interest [5]. Healthcare providers for compliance are to implement a WMP as the recommended process for mitigating risk. The building owner is directed to establish a WMP in accordance with the American National Standards Institute (ANSI) and American Society of Heating, Refrigerating and Air–Conditioning Engineers (ASHRAE) Standard 188 *Legionellosis*: *Risk Management for Building Water Systems* and the *CDC Toolkit Developing a Water Management Program to Reduce Legionella Growth* & *Spread in Buildings*. Both documents identify construction as a known risk factor associated with the growth and spread of waterborne diseases and in specific legionellosis [3,7]. ANSI/ASHRAE Standard 188–2018 Section 4.2.1 Building Owner Requirements states the building owner is to survey each existing building prior to renovation, addition or modification and any new building prior to occupancy [7]. Although the WMP survey process is adequately described, the ANSI/ASHRAE Standard 188 lacks description of the specific construction risk factors to assist the building owner with implementation of a WMP during demolition, renovation, additions or new construction. The CDC Toolkit suggests construction activities related to vibration, water pressure changes dislodging biofilms and soil and sediment invasion from water main breaks influence waterborne pathogen growth and spread [3].

These construction activity descriptions, however, lack characterization of risk. In order for a building owner’s water management team to navigate a successful risk management process, the team members must understand the frequency of construction activities and clinical severity to determine risk. A similar risk characterization is commonly performed in healthcare settings for airborne pathogens [8]. Risk management for building water systems is a series of systematic steps to reduce risk and is an established process for controlling waterborne pathogens [2,3,7,9]. A risk is the potential to harm humans from exposure (in this context) to a waterborne pathogen [10,11]. Risk is a product of frequency times severity expressed in a formula as: Risk = frequency × severity

Frequency is a measure of probability. Severity is a measure of consequence. As frequency and/or severity increase, risk increases. Assessing and managing risk is an essential element of environmental, clinical and public health practice for implementing a WMP program.

The focus of this study is to integrate evidence to categorize and define severity for construction activity risk factors associated with waterborne pathogens in community and healthcare settings. The present study aims are: (1) assess populations at risk and identify environmental conditions contributing to the problem; (2) categorize construction activity risk factors; (3) quantify severity for each construction activity risk factor by health effect (i.e., the number of disease cases and deaths); and (4) identify current knowledge gaps relating to construction in water management programs. The present study findings are intended to assist WMP team members (building owners, facility managers, construction managers, infection preventionists, risk managers and safety officers) to implement an ongoing comprehensive WMP inclusive of construction and commissioning activities.

## 2. Materials and Methods

The Preferred Reporting Items for Systematic Reviews and Meta-Analyses (PRISMA) [12] in conjunction with Rooney et al. [13] was the framework for evidence integration of a literature-based environmental health assessment. Rooney et al. [13] was selected for Steps 4 through 7 [13] (p.712) to assess the quality of individual studies and rate the confidence in the body of evidence (BOE). A search of scientific literature was performed covering multiple databases (CINAHL, Embase, MedLine, PubMed and Web of Knowledge with Web of Science, as well as Cochrane Central). Search terms took into account the environmental setting (community or healthcare); the waterborne pathogens of interest from the CMS S&C 17–30 Memo [5] [*Legionella*, *Pseudomonas*, *Acinetobacter*, *Burkholderia*, *Stenotrophomonas*, NTM and fungi (*Aspergillus* and *Fusarium*)]; and construction types (new construction, renovation, additions, expansions or demolition). A Boolean phrase template was constructed and terms inserted into each database as follows: ((environmental setting)) AND ((pathogen)) AND ((water)) AND ((construction type)). The search was conducted in October 2018, with no restriction on date of event or geographic location.

### 2.1. Review Process

The review process was completed in three steps: identification, screening and eligibility [12]. Identification involved reviewing the articles output from each database and downloading all relevant titles into a web-based citation manager (RefWorks Classic Version). Duplicates were removed. Titles and abstracts were reviewed to determine the number of full text articles eligible for assessment.

### 2.2. Inclusion and Exclusion Criteria

All articles reporting disease cases or deaths suspected to be associated with construction activities and a waterborne pathogen were included. The study inclusion parameters were: (1) human disease cases suspected to be associated with construction activities including renovation, additions, demolition or new construction; and (2) the pathway of exposure to the human host involved water reservoirs related to construction activities or building water systems. The review was inclusive of both community-acquired and healthcare-associated disease cases. All studies in the review were available through a R1 university database search engine and published in peer-review journals as research articles, conference proceedings or poster presentations.

Articles were excluded based upon construction activities associated with: (1) surface transmission of mold from water damaged building materials; (2) environmental settings concerning disaster relief, waste water treatment, animal farms, dams or municipal water stations; or (3) airborne pathogens without a water source. Additionally, article exclusions were: (4) republishing the same event under another author’s name except for an article in English originally published in Spanish [14]; (5) existing building studies about waterborne pathogens without mention of construction activities; (6) mentioning construction activities as a general risk; and (7) using construction terms to describe other research topics (see Figure 1).

### 2.3. Data Extraction

Relevant data from the remaining full-text articles were identified and extracted into a health system matrix [15] of information including key study characteristics related to the: outbreak event, demographics of the populations at risk, environmental and building conditions and construction activities. A meta-analysis was not performed due to the heterogeneity of the methods and measures reported in each article.

### 2.4. Assessing Risk of Bias

Articles were assessed using study quality queries from Rooney et al. [13] for risk of bias for the applicable study designs (case-control studies, case series, prospective or retrospective cohort studies, cross-sectional studies and case reports). Two reviewers independently rated each article to determine risk of bias in each article as: definitely low, probably low, probably high or definitely high risk of bias.

### 2.5. Determining Risk Severity Using Confidence in the Body of Evidence

Each reviewer completed a code sheet and assigned a BOE rating (high, moderate, low or very low). Very low BOE ratings did not advance to the final stage of quantitative analysis. Using the known formula for risk, reviewers determined a value for severity by correlating construction activity risk factors with the number of disease cases and deaths reported across all studies. 

## 3. Results

The present study yielded 31 articles [16,17,18,19,20,21,22,23,24,25,26,27,28,29,30,31,32,33,34,35,36,37,38,39,40,41,42,43,44,45,46] meeting the eligibility criteria for quantitative review and synthesis (see Table 1). These studies cover construction activities associated with waterborne disease cases from 1965 to 2016 (i.e., publication dates were from 1978–2018). A total of 894 disease cases were reported inclusive of 112 deaths. The mean event duration (from index case to final case) was 13.8 months (n = 31). Community event mean duration was 1.9 months, while healthcare event mean duration was 17.3 months. The mean number of disease cases and deaths per event respectively were: 28.8/3.6 (total, n = 31); 36.7/1.4 (community, n = 7) and 26.5/4.3 (healthcare, n = 24). The waterborne pathogen most reported in association with construction activities was *Legionella*, *spp.* (n = 26, 83.9%) followed by NTM (n = 3, 9.7%), *Fusarium* (n = 1, 3.2%) and *Sphingomonas* (n = 1, 3.2%). *Pseudomonas*, *Burkholderia*, *Stenotrophomonas* and *Aspergillus* did not appear as pathogens of interest based on the criteria set forth in the present study. *Acinetobacter* appeared in one study based on inclusion criteria, however it was ultimately excluded based on high risk of bias and very low confidence in the body of evidence [47]. All outcome data are reported for all studies (n = 31), as well as the two major categories defining the environmental setting for disease cases: community-acquired (n = 7) or healthcare-associated (n = 24) (see Table 2).

### 3.1. Demographic Characteristics

The demographics associated with waterborne pathogens and construction activities are similar to the demographics for disease cases associated with waterborne pathogens in existing building water systems. Common characteristics for disease cases from waterborne pathogens in existing building water systems not undergoing construction are known to be primarily male, >50 years of age, smokers and a person with immunocompromised health status [4,48]. In the present study demographic characteristics of the populations at risk were not consistently reported across all studies for sex, age, smoking status and underlying disease status. When demographic characteristics were reported (see Table 3), the present study found: more males (N = 362) than females (N = 193); a mean age >50 years (mean age = 58.4 years) and studies identifying disease cases with underlying disease status (48%, n = 15). Patients receiving medical care at a healthcare facility (N = 625, 70.0%) were the population most at risk. Although healthcare workers and visitors to healthcare settings were also reported, these were <1% of the total population at risk. Community-acquired disease cases represented 27.7% (N = 248) of the population, while construction workers were identified as 1.5% (N = 13) of the population at risk.

### 3.2. Environmental Characteristics

The characteristics of the environmental setting most associated with waterborne pathogen growth and spread for construction projects involved: USA geographic locations, acute care hospitals, addition/expansions on an existing campus, fall and winter months and the potable building water distribution system to sinks and shower fixtures (See Table 4).

#### 3.2.1. Geographic Location

The geographic location of disease cases associated with construction activities and waterborne pathogens were most reported in the USA. Nineteen studies were located in the USA including: Alabama (1), California (2), Connecticut (1), District of Columbia (2), Illinois (1), Iowa (1), Maryland (1), Minnesota (1), North Carolina (1), Ohio (1), Pennsylvania (3), Rhode Island (2) and Texas (2). These states represent a concentration of disease cases in the Northeast and Midwest of the USA, as well as urban states with larger populations such as California and Texas. Other North American studies were from disease cases in the Canadian provinces of Alberta (1), British Columbia (1), Ontario (1) and Quebec (1). The remaining international waterborne disease cases associated with construction activities were from: United Kingdom (2), Spain (3), France (1), Australia (1) and Italy (1).

#### 3.2.2. Seasonality and Weather Conditions

Construction activity risk factors need to be considered a year-round impact and not predominately occurring within a summer season condition. Warmer seasonal weather creates warmer baseline water conditions typically enhancing growth and spread of waterborne pathogens [48]. The calendar season in which disease cases associated with construction activities occurred was greatest during fall and winter seasons. The present study found 54% of the events associated with construction activities occurred in fall and winter seasons, while 46% occurred in spring and summer months. For example, Knox et al. [43] noted the anomaly of having a legionellosis outbreak during a sub-zero Canadian cold climate condition. Miragliotta et al. [44] reported two construction workers contracted legionellosis while building an artesian well in Italy. The exposure to excavation and water emerging from a spraying wellhead jet occurred in underground cooler conditions. In addition to seasonality, in the present study weather conditions amplified construction activity risk factors in 32% (n = 10) of the studies. When weather was a factor, four conditions were identified: prevailing winds (80%), temperature (50%), humidity (30%) and heavy rain or storms (20%).

#### 3.2.3. Building Types

The building type for disease cases associated with construction activities and waterborne pathogens were greater in healthcare (77%) than community (23%) building settings. Seventy-one percent (71%) of the healthcare settings involved acute care hospital operations. A smaller percent represented long-term care (3%) and psychiatric/behavior health (3%) building types. For community settings 13% involved a non-healthcare building (industrial, retail and medical school construction) An additional 10% were from construction activities for open land development concentrated in a geographic region. Castilla et al. [40] reported extensive construction activities overburdening cooling towers in the Pamplona region of Spain. Even with a cooling tower registration program, the local public health officials were not aware of the geographic concentration of construction activities generating airborne debris which settled into cooling towers. The cooling towers were overburdened and produced drift containing *Legionella*.

#### 3.2.4. Construction Types

Disease cases emerge during construction activities associated with waterborne pathogens when construction is occurring within, around or adjacent to healthcare settings. Twenty-three studies (74%) reported disease cases and deaths around active patient care zones. Construction work performed where disease cases emerged involved a full range of activities such as renovation, additions, new campus buildings or demolition activities to remove older buildings. Demirjian et al. [21] described 22 LD cases emerging over a two-year period after an extensive period of potable water distribution system renovations occurred. Similarly, an outbreak of 56 LD cases were reported in a neighborhood where a medical school was being constructed near a health clinic in the community district in Barcelona, Spain [41].

#### 3.2.5. Water Reservoirs

Construction activities impacted the potable water distribution system 48.4% (n = 15). Utility water systems related solely to cooling towers led to disease cases 16% (n = 5) of the time. Multiple building water systems were associated with disease cases 13% (n = 4) of the time. Ten percent (n = 3) of studies reported propagation of an airborne pathogen into a water source (e.g., weather condition or water reservoir). The pathogen traveled, similar to drift from a cooling tower, over a geographic area resulting in an increased number of disease cases [40].

### 3.3. Construction Activity Risk Factors and Severity

Each study reported one or more construction activities contributing to disease cases and deaths. The nine construction activities by frequency of occurrence reported were: excavation (38.7%, n = 12); inadequate building water system commissioning strategies at beneficial occupancy and building opening (35.5%, n = 11); re-pressurization (29.0%, n = 9); demolition activities (19.4%, n = 6); water efficiency challenges at building opening (16.1%, n = 5); construction equipment with a water reservoir (9.7%, n = 3); vibration activities (9.7%, n = 3); underground utility connections (9.7%, n = 3); and water main breaks (3.2%, n = 1).

#### Risk Severity

The construction activity associated with the most waterborne disease cases and deaths was inadequate commissioning of the building during beneficial occupancy (i.e., while preparing for the building opening to the public). Each construction activity was assigned a severity (S) rating based upon correlation to total disease cases (S1) and deaths (S2). The assessment process for risk of bias and confidence in BOE (see Table 1) revealed 20 articles with high confidence, nine articles with moderate confidence, two articles with low confidence and four articles with very low confidence. Thirty-one articles (88.6%) achieved a BOE rating high enough to correlate a construction activity with a health effect. The four articles with very low BOE rating did not qualify for further assessment [47,49,50,51]. The remaining 31 articles were integrated to establish a value for severity of each construction activity correlated with the two health effects (See Table 5).

## 4. Discussion

The present study revealed waterborne pathogens were associated with one or more construction activity risk factors in 31 studies about events occurring from July 1965 to December 2016. Many of these disease cases (N = 894) and deaths (N = 112) were preventable. Most of these disease cases (N = 637, 71.3%) emerged during a patient stay in a healthcare setting. These 31 studies most likely represent a small segment of the actual disease cases and deaths. More events like these likely go unidentified, are not published in scientific literature or (other than legionellosis) do not involve mandatory reporting to public health agencies [4]. In addition to risk factor identification and ranking, the present study findings indicate five knowledge gaps about construction activity risk factors associated with waterborne pathogens which need illumination to reduce illness and death in community and healthcare settings. Water management teams will need to evaluate these knowledge gaps and translate these findings to develop a more robust water management program.

### 4.1. Gap #1: Historical Significance of Construction Activities Impacting Waterborne Pathogens

Soil and debris from construction activities are part of the historical significance of waterborne pathogens affecting community-acquired and hospital-associated disease cases. In 1954, Drozanski, a Polish scientist, first isolated *Sacrobium lyticum* from soil amoeba [48,52,53]. In doing so, Drozanski established the presence of an intracellular microorganism which lysed amoebae due to bacterial infection. Lysis is the process of a cell wall or membrane disintegrating and ultimately rupturing the cell. When soil containing pathogens enters water reservoirs, bacteria can proliferate and develop into biofilm. Drozanski’s *Sacrobium lyticum* was reclassified with the genus *Legionella* after the pathogen was named related to the LD outbreak at the American Legion Conference in 1976 at the Bellevue-Stratford Hotel, Philadelphia, Pennsylvania. Drozanski’s seminal research was the connection of soil specifically to *Legionella* and how other opportunistic waterborne pathogens survive to become resistant to disinfection treatment in water reservoirs.

In many building projects construction activities involve direct contact with soil and sediment. The present study found construction activities involving connections between soil intensive activities resulted in 407 disease cases and 48 deaths. In 1965, an outbreak at St. Elizabeth’s Hospital in Washington, DC was associated with construction activities for excavation of an underground landscape irrigation system [38]. Thacker et al. [38] in 1978 utilized samples collected and stored by the CDC since 1965 to determine this outbreak (81 disease cases and 14 deaths) was connected to the *Legionella* pathogen. Using fluorescent antibody testing the investigators identified the *Legionella* pathogen. Thacker et al. [38] reported statistically significant correlations to patients who developed a disease case with proximity closer to construction excavation. Patients who slept closest to open windows (P < 0.01) and those with ground privileges (P < 0.0001) were more likely to have been a disease case. This event marked the first USA hospital-associated LD cases and deaths.

Additionally, Pontiac Fever was first described in 1968 which was later linked to *Legionella, pneumophila* [54]. The events in Pontiac, Michigan are also believed to have connections to construction activity risk factors. Glick et al. [54] examined the etiology of the Pontiac County Health Department’s “explosive epidemic” (p.149). This event involved 144 symptomatic cases of employees and visitors occupying the building. An epidemiology investigation reported the likely source of the outbreak to be a multi-chambered air-conditioning system with an open reservoir in the evaporative condenser. The condenser sprayed water into the metal duct system to combine with air for cooling the building. The year following the outbreak investigation the air conditioning system was rebuilt and no further symptomatic cases emerged. Glick et al. [54] acknowledged some foreign particulate matter most likely entered into the air system triggering growth and spread of an air/waterborne pathogen. The investigators also noted the facility itself was not under renovation. However, in early June 1968 immediately prior to the disease cases emerging, the ground areas adjacent to the building were cleared and paved “raising clouds of dust that at times enveloped the building” [54] (p.150). Heavy rains and increased temperatures occurred the week before the outbreak began.

Comparing the 1968 Pontiac County Health Department’s construction events to the present study findings suggests water reservoirs may have become compromised from two construction activity risk factors. First, soil and debris from demolition activities may have entered the mechanical system. By example, Monforte et al. [14] described excessive soil and debris from localized construction sites spread across a multiblock zone of Barcelona, Spain. An LD outbreak ensued with 56 disease cases and 7 deaths. Second, a water reservoir source within construction equipment used for paving may have contributed to the Pontiac outbreak. Coscolla et al. [42] reported a community outbreak of LD from a milling machine used in street asphalt paving. A water reservoir filled from an industrial non-potable water source became stagnant and proliferated with *Legionella*. The outbreak of 11 LD cases was traced to the areas where the milling machine was used in a region of Alcoi, Spain.

The National Academy of Sciences, Engineering and Medicine report on Managing *Legionella* in Water Systems contains a timeline related to the historical significance and retrospective analysis of *Legionella* outbreaks [53]. Drozanski 1954, St. Elizabeth’s Hospital 1965 and Pontiac Fever 1968 (all related to soil and construction activities) are identified as having played a role in the evolutionary knowledge about *Legionella*. Yet, these events and their association with construction activity risk factors is not mentioned.

### 4.2. Gap #2: Updating Water Management Standards with Construction Activity Risk Factors

The model standards and guidelines for building water management programs need to be updated to include construction activity risk factors. The standards and guidelines clearly identify construction as a known risk impacting a building water management program [3,7]. Yet, these same standards lack specificity about construction. Without clear descriptors for construction risk factors, building owner compliance remains challenging. ANSI/ASHRAE Standard 188 mentions planned and unplanned water service disruptions reducing water pressure below 20 psi (140 KPa) potentially connected to new construction tie-ins, other systems of repair and emergency conditions [7]. But the construction risk factors associated with these situations are not listed. By comparison, the CDC Toolkit does mention four construction risk factors which align with the present study findings (excavation, re-pressurization, vibration activities and water main breaks) [3]. The CDC Toolkit also mentions unoccupied areas, renovations causing a dead-end conditions or reduced occupancy in hotels during seasonal fluctuations leading to water stagnation. The present study identified five additional construction activity risk factors: demolition (debris and sediment), commissioning activities at building opening and occupancy, water efficiency challenges at building opening, underground utility connections and construction equipment using water reservoirs. Industry standards and best practice guidelines should consider identifying construction risk factors in a dedicated section for building owners to address water management before, during and after construction activities. Public health policy initiatives should consider updated language emphasizing awareness of construction activities near cooling towers as a possible source of waterborne pathogens in community settings. In healthcare settings, risk management for building water systems needs to emphasize high risk factors for construction activities at the beginning of the construction project (underground activities) and at the end of the construction project (commissioning activities).

### 4.3. Gap #3: Establish Water Management for Construction & Commissioning Plans

Water management programs must contain provisions for construction and commissioning plans to reduce the impact of waterborne pathogens before, during and after construction activities. Only 7 studies (22.6%) reported implementing some aspect of a water management plan during construction activities. Additionally, 21 studies (67.7%) did not mention having any hazard controls in-place prior to starting construction activities. After disease cases emerged, 23 studies (74.2%) identified hazard controls were necessary to bring an outbreak to an end. These findings demonstrate a reactionary approach to dealing with construction activity impacts to the building water systems. This finding is similar to the CDC’s findings for environmental control deficiencies during LD outbreaks [4]. By categorizing construction activity risk factors and establishing a value for risk severity, building owners and their respective water management team members can characterize construction risk factors when developing a water management plan for construction. The present study noted two types of risk severity. Severity 1 is the number of disease cases and Severity 2 is the total number of deaths associated with each construction activity risk factor (see Table 5).

In addition to the number of disease cases and deaths, other present study findings should increase concern about construction activities contributing to the growth and spread of waterborne pathogens. First, populations exposed to a waterborne pathogen associated with construction activities are not sporadic, but rather outbreak intensive. In the present study, the mean number of disease cases reported across all 31 studies was 28.8 cases, inclusive of 3.8 deaths per event. Community-acquired disease cases from water reservoirs with high numbers of influenza reported cases were quickly addressed while seeking out offending water sources. However, in healthcare settings the primary source of a rapidly emerging disease cases became confused around a myriad of other circumstances. In Garbe et al. [22] the outbreak investigation spent 60 days looking at the potable water distribution system after a second patient tested positive for LD. Fifteen disease cases inclusive of 10 deaths emerged, before realizing a new cooling tower was improperly commissioned. Drift emitting from a new addition roof top cooling tower circulated into an existing adjacent patient bed tower. Second, investigators expressed lack of knowledge about construction worker exposure especially at healthcare construction sites [16]. Two Canadian construction workers renovating a hospital roof area checked into the same hospital with severe flu symptoms. Both developed into LD cases. A subsequent female LD patient case was identified in a room located 2 stories above the same construction area. Three cases emerging within a short time frame created confusion as to the origin of the waterborne pathogen. A construction worker clinical sample and a condensing tower environmental sample had the same polymerase chain reaction (PCR) sequence linking the two samples. The female patient case was related to a positive cold-water sample from a tap in the patient room. Third, infection preventionists and healthcare leadership teams were baffled to be investigating a waterborne pathogen issue in a newly constructed area or building [17,22,24,39]. Baker et al. [17] reported this type of incident in a two-phase outbreak involving *Mycobacterium abscessus.* The potable water system used in pulmonary treatments and aerosols generated from colonized cardiology heater cooler units resulted in 116 disease cases inclusive of 26 deaths over a 36-month period. In contrast Prabaker et al. [33] described discussions with physician colleagues at another facility about their experience with rapidly growing *Mycobacterium, gordonae* in a newly constructed hospital building. These discussions helped avoid disease cases at another facility. Lastly, unoccupied spaces due to delays in building activation were also problematic [27,39]. Johnson et al. [27] stated a one-year post-construction building start-up led to managing a long-term waterborne pathogen with emerging cases over the next decade.

### 4.4. Gap #4: Moving Beyond Legionella

Although *Legionella* disease cases were reported in connection with construction activity risk factors, additional waterborne pathogens need to be recognized as equally challenging under such conditions. The present study found 697 (78%) disease cases and 83 (74%) deaths were associated with *Legionella*, *spp*. [16,18,19,20,21,22,23,25,26,29,30,31,32,35,36,37,38,39,40,41,42,43,44,45,46]. Other opportunistic waterborne pathogens such as NTM and *Sphingomonas*, *spp.* accounted for 197 (22%) disease cases and 29 (26%) deaths [17,24,33]. Investigators conducting studies about non-*Legionella* waterborne pathogens find minimal emphasis is placed on water management across all microbials (i.e., all genera/species) [24,27,55]. Guspiel et al. [24] complemented ANSI/ASHRAE Standard 188 committee for establishing minimum standard of care document and raising awareness within the administrative C-Suite. However, an acute emphasis on *Legionella* marginalizes other infectious waterborne disease issues. Guspiel et al. [24] dealt with an outbreak condition of rapidly growing mycobacteria when pediatric stem cell transplant patients were moved into a new children’s hospital. Oversized ice machines selected for water efficiency had to be replaced with smaller units and remediation of all drinking fountains and ice machines was necessary. For 10 months after building opening, 15 disease cases persisted when there had been zero transplant pediatric patient infections the prior year in the older building. This outbreak experience led investigators to express collaboration efforts to move beyond *Legionella* awareness are essential. Johnson et al. [27] emphasized the CMS S&C 17–30 Memo requirements for healthcare facilities to establish water management programs referencing an expanded list of waterborne pathogens. While an epidemiologic investigation is encouraged by the CDC for a single case of legionellosis, yet other more plaguing waterborne pathogens could settle into an institution’s building water system, go unrecognized, remain underreported and proliferate over a long period of time.

### 4.5. Gap #5 Education & Training Building Design and Construction Industries

The design and construction industry must obtain additional training and education in water management for construction and commissioning to assure building water systems are safe for human occupancy. Water quality during construction focuses primarily on the Environmental Protection Agency’s (EPA) Safe Drinking Water Act (SDWA) which responds to chemical and microbial contaminants related to humans ingesting water (drinking through the mouth) [56]. The model plumbing codes require disinfection and flushing of the underground building main and the potable water distribution systems during the process of building assembly [57]. Nevertheless, assuring water is safe post-construction for inhalation (breathing aerosolized water vapor or droplets into the lungs) or water surface transmission to minimize hospital-acquired infections has never been standardized. When ANSI/ASHRAE Standard 188 [7] was published, its primary audience was the building owner [58]. Yet, several construction and commissioning activities mentioned in the Standard would typically require the involvement of design and construction professionals to lead these efforts. The challenge is building design and construction professionals are less familiar with water management, water chemistry, microbiology, industrial hygiene or medicine-related fields to understand the transmission of disease from waterborne pathogens [53]. The National Academy of Sciences Engineering and Medicine (NASEM) note these design and construction industries are greatly impacting the public water system [53]. NASEM suggested training is paramount among these professionals to balance design, construction and ongoing operations of building water systems with an equivalent base of knowledge by others in water management.

Additionally, green building design and construction professionals must address perpetuating the concept of water efficiency without any consideration for water safety. Since the inception of building rating systems for energy efficiency, design professionals in conjunction with building owners strive for energy cost savings related to water [59]. Also, consumer spending has notably increased toward environmentally conscious businesses [60]. Unfortunately, water efficiency objectives increase water age in building distribution systems impacting overall water safety [53,59]. The present study findings indicated water efficiency devices within the building water system need to be properly commissioned. At building start-up facility maintenance personnel were unaware to check for adequate temperature levels [27,36,37], water flow [17,24,27], malfunctioning components [37] and improperly sized equipment [24,37] to avoid negatively impacting building water systems. Water efficiency challenges at building start-up were found in the present study to contribute to 168 disease cases and 32 deaths. Baker et al. [17] reported a water efficiency standard led to high water age in the building’s water distribution system. Low flow fixtures, hot water recirculating loops and prolonged hot water storage reduced the amount of fresh water entering the building water system. The water disinfection parameters were reduced leaving patients vulnerable to an under protected water supply. Baker et al. [17] recommended any healthcare facility with high-efficiency water standards will need to consider periodic water system flushing to reduce stagnant conditions, as well as removal of water flow restrictors (i.e., aerators and laminar flow devices) and minimize redundancies. In 2016 Rhodes et al. [59] study on green water efficiency concluded chlorine and chloramine residuals in green building water systems experienced a high rate of disinfection residual decay (20–144 times greater) than the control–a water sample in a glass container from the same building water system sitting for identical time periods. The interdependency of water safety and water efficiency calls into question the chemical and microbial reliability of building water systems with high water age which were designed solely around water efficiency standards.

### 4.6. Study Limitations

The present study limitations include not evaluating events included in the press or industry articles or Internet reports from government organizations or agencies. All studies included were reported in peer-review literature and were an event involving a waterborne pathogen and construction activities. Additionally, there is a potential for publication or editorial review bias on articles about *Legionella.* Studies about waterborne pathogens and construction centered around two time periods. In the first decade after the 1976 LD event in Philadelphia, several investigators associated construction activities with *Legionella* outbreaks. The topic remained infrequently reported until the CDC published concerns about the increased number of LD cases from 2000 to 2014. This may have impacted the number of articles focusing on *Legionella* versus reporting on other pathogens of interest.

## 5. Conclusions

In the present study 31 studies reported waterborne pathogens associated with construction activity risk factors affected 894 people who developed a serious health effect from July 1965 to December 2016. The number of disease cases and deaths are undoubtedly under reported in this time period due to under reporting of events and mis-diagnosed disease cases. The present study categorized nine construction activity risk factors which contributed to the growth and spread of waterborne pathogens in both community and healthcare settings. Integrating the construction activity risk factors with the five knowledge gaps is essential for building owner compliance with standards requiring water management during construction as part of implementing an ongoing comprehensive water management program. Current risk management practices for water management do not illuminate on the specific construction activity risk factors, nor establish a severity of consequence for such activities. Many of the disease cases represented in the present study appear to be preventable moving forward, if the following measures are taken:(1)building owners maintain a water management program inclusive of developing a water management for construction (WMC) plan before construction activities begin;(2)building owners take responsibility to assure a proper building water management commissioning plan is implemented prior to occupancy of any building (e.g., newly constructed, renovated, unoccupied and re-opened, change of use or occupancy and building acquisition) with an emphasis on healthcare settings for acute patient care services;(3)public health agencies monitor cooling tower chemical treatment parameters in community settings, which can become overburdened with airborne soil and sediment from nearby construction activities;(4)building owners, public health officials, facility managers, facility construction managers, infectious disease physicians, infection prevention specialists and design and construction professionals obtain proper training, education and awareness of construction activity risk factors as described in the present study;(5)facility construction managers, architects, engineers and construction professionals develop and implement better evidenced-based construction policies and communication strategies to reduce the impact of the nine construction activity risk factors on building water distribution systems particularly in healthcare settings.

More research findings about the connection between opportunistic waterborne pathogens and construction activities are needed to improve the generalizability of the scientific findings on this research topic. The present study findings support a comprehensive building water management program inclusive of construction activity risk factors for renovation, additions, demolition and new construction projects. Standards, codes and guideline documents addressing risk management for building water systems need to be updated and inclusive of all known construction activity risk factors. Waterborne disease outbreaks are associated with construction activities in both community and healthcare settings. However, healthcare facilities are more likely to have a vulnerable patient population impacted by such an exposure. A building owner can no longer assume these construction activity risk factors associated with waterborne pathogen disease cases are insignificant or accidental.

## Figures and Tables

**Figure 1 ijerph-17-02168-f001:**
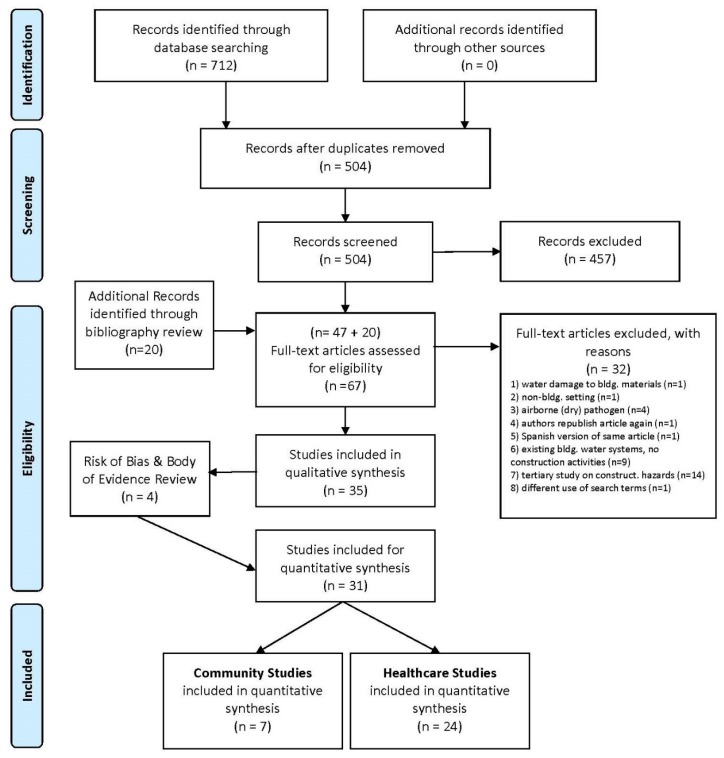
Article selection process [12].

**Table 1 ijerph-17-02168-t001:** Article/study characteristics and quality assessment of articles.

Ref.	Authors	Year Pub.	Geographic Locations Country, Region	BOE Rating	Year Event Began	Event Duration (Months)	Pathogen of Interest	Total Disease Cases	Total Deaths	Construction Risk Factor(s)
**Healthcare-associated**										
[14]	Abbas, et al.	2003	CAN, ON	3	2002	2	*Legionella*	5	0	Demolition, repressurization
[15]	Baker, et al.	2017	US, NC	3	2013	29	*Mycobacterium*	116	26	Commissioning at building opening, and water efficiency challenges
[16]	Blatt, et al.	1993	US, TX	2	1989	12	*Legionella*	14	6	Excavation, underground utility connections
[17]	Boivin, et al.	2012	CAN, QC	1	2008	3	*Legionella*	2	0	Excavation, vibration
[18]	Chafin, et al.	2011	US, TX	2	2006	3	*Legionella*	10	0	Water main, repressurization
[19]	Demirjian, et al.	2015	US, PA	3	2011	24	*Legionella*	22	6	Repressurization
[20]	Garbe, et al.	1985	US, RI	3	1983	3	*Legionella*	15	10	Repressurization
[21]	Grove, et al.	2002	AU, Adelaide	3	2000	5	*Legionella*	7	2	Demolition
[22]	Guspiel, et al.	2017	US, MN	3	2011	10	*Mycobacterium*	15	0	Commissioning at building opening, water efficiency challenges
[23]	Haley, et al.	1979	US, CA	3	1977	15	*Legionella*	49	15	Excavation, underground utility connections, commissioning at building opening
[24]	Helms, et al.	1983	US, IW	3	1981	10	*Legionella*	24	11	Commissioning at building opening
[25]	Johnson, et al.	2018	US, DC	3	2005	144	*Shingomonas*	31	3	Commissioning at building opening, water efficiency challenges
[26]	Kandiah, et al.	2012	US, PA	2	2011	9	*Legionella*	0	0	Repressurization
[27]	Marks, et al.	1979	US, OH	1	1977	4	*Legionella*	9	0	Excavation
[28]	Martin, et al.	1988	CAN, NS	2	1984	2	*Legionella*	8	2	Demolition, excavation
[29]	Mermel, et al.	1995	US, RI	2	1992	3	*Legionella*	2	2	Repressurization, vibration
[30]	Parry, et al.	1985	US, CT	3	1983	5	*Legionella*	5	0	Excavation, underground utility connections, demolition, repressurization
[31]	Prabaker, et al.	2015	US, IL	3	2012	12	*Mycobacterium*	35	0	Commissioning at building opening
[32]	Sautour, et al.	2012	FR, Dijon	2	2009	9	*Fusarium*	0	0	Repressurization, vibration
[33]	Shands, et al.	1985	US, CA	3	1978	47	*Legionella*	171	0	Commissioning at building opening, repressurization
[34]	Srivastava, et al.	2011	UK	3	2007	22	*Legionella*	0	0	Commissioning at building opening, water efficiency challenges
[35]	Stout, et al.	2000	US, PA	3	1992	36	*Legionella*	6	3	Commissioning at building opening, water efficiency challenges
[36]	Thacker, et al.	1978	US, DC	3	1965	3	*Legionella*	81	16	Excavation
[37]	Watkins, et al.	2017	US, AL	3	2014	4	*Legionella*	10	0	Commissioning at building opening
**Community-acquired**										
[38]	Castilla, *et al.*	2008	ES, Pamplona	3	2006	2	*Legionella*	146	0	Demolition, excavation
[39]	Cayla, *et al.*	1989	ES, Barcelona	3	1988	2	*Legionella*	56	7	Demolition, excavation
[40]	Coscolla, *et al.*	2010	ES, Alcoi	3	2009	3	*Legionella*	11	0	Construction equipment w/water
[41]	Knox, *et al.*	2017	CAN, AB	2	2012	2	*Legionella*	8	0	Construction equipment w/water, excavation
[42]	Miragliotta, *et al.*	1992	IT, Apulia	2	1990	1	*Legionella*	2	0	Construction equipment w/water, excavation
[43]	Morton, *et al.*	1986	UK, Lancaster	2	1981	2	*Legionella*	7	1	Did not determine source
[44]	Redd, *et al.*	1990	US, MD	3	1986	1	*Legionella*	27	2	Excavation

Body of evidence (BOE).

**Table 2 ijerph-17-02168-t002:** Event characteristics.

Category	All	Community	Hospital
	n = 31	n = 7	n = 24
**Location**			
US	19	1	18
Other International	6	4	2
Canada	4	1	3
UK	2	1	1
**Waterborne Pathogen of Interest**			
*Legionella*, *pneumophila serogroup 1*	19	6	13
*Legionella*, *other species*	2	0	2
*Legionella*, *no species listed*	5	1	4
Nontubuculous Mycobacteria	3	0	3
*Fusarium*	1	0	1
*Sphingomonas*	1	0	1
**Disease Cases & Deaths**			
Mean (cases per event)	28.8	36.7	26.5
Mean (deaths per event)	3.6	1.4	4.3
Total cases	894	257	637
Confirmed	794	247	547
Probable	76	10	66
Suspected	24	0	24
Deaths	112	10	102
**Event Duration**			
Mean (months)	13.8	1.9	17.3
0 to 6 months	18	7	11
>6 months to 12 months	6	0	6
>12 months to 18 months	1	0	1
>18 months to 24 months	2	0	2
>24 months	4	0	4

**Table 3 ijerph-17-02168-t003:** Demographic characteristics.

Category	All	Community	Hospital
	n = 31	n = 7	n = 24
**Total Population**	894	257	637
Patients	625	0	625
Community residents	248	248	0
Construction workers	13	9	4
Healthcare staff	4	0	4
Visitors	4	0	4
**Sex**	(n = 19)	(n = 7)	(n = 11)
Males	362	160	202
Females	193	97	96
**Age**	(n = 20)	(n = 7)	(n = 13)
Range for low age	10 yrs–71 yrs	21 yrs–51 yrs	10 yrs–71 yrs
Range for high age	38 yrs–97 yrs	38 yrs–97 yrs	47 yrs–87 yrs
**Age Mean**	(n = 21)	(n = 7)	(n = 14)
	58.4 yrs	55.3 yrs	60.0 yrs
**≥50% Population w/Underlying Disease**			
Yes	15	1	14
No	3	2	1
Did not report	13	4	9
**≥50% Population Smoked**			
Yes	6	2	4
No	2	0	2
Did not report	23	5	18

**Table 4 ijerph-17-02168-t004:** Environmental and building characteristics.

Category	All	Community	Hospital
	n = 31	n = 7	n = 24
**Building Type**			
Healthcare, acute care hospital	22	0	22
Open Land Development	3	3	0
Industrial	2	2	0
Healthcare, long term care	1	0	1
Healthcare, psych/behavioral	1	0	1
Medical school	1	1	0
Retail	1	1	0
**Construction Type**			
Addition/Expansion on-site	12	1	11
Demolition only	7	3	4
New Construction off-site	5	2	3
Renovation within existing building	3	0	3
Renovation + New Construction	3	0	3
**Season**			
Winter	11	1	10
Spring	7	2	5
Summer	7	1	6
Fall	6	3	3
**Weather Impacts**			
Yes	10	7	3
No	21	0	21
**Weather Factors**			
Prevailing winds	8	5	3
Temperature	5	4	1
Humidity	3	3	0
Rains/Storms	2	1	1
**Building Water System Impacted**			
Potable water system	15	0	15
Utility (Cooling Towers)	5	2	3
Multiple systems impacted	4	1	3
Airborne with moisture source	3	2	1
Non-potable water system	2	2	0
Undetermined	2	0	2
**Water Reservoir Impacted**			
Building water piping distribution system	17	0	17
Sink or shower fixture	10	0	10
Cooling dower	7	2	5
Dead-Leg or deadend piping condition	4	0	4
Construction equipment	3	3	0
Ice machine	3	0	3
AC Unit	3	1	2
Filtration system	2	1	1
Irrigation system	2	0	2
Recirculation pump	2	0	2
Standing water (ponding)	1	0	1
Did Not Report	1	0	1

**Table 5 ijerph-17-02168-t005:** Construction activity risk factors & severity rank.

Construction Activity Risk Factor	Frequency Mentioned	Total Disease Cases (S1)	Total Deaths (S2)	Severity Rank (S1)	Severity Rank (S2)
Commissioning @ Building Occup./Opening	11	472	68	1	1
Excavation	12	407	48	2	2
Re-pressurization (Shut-downs & Start-ups)	9	230	18	3	5
Demolition Activities	6	227	11	4	6
Efficiency Design Challenges	5	168	32	5	3
Underground Utility Connections	3	68	21	6	4
Construction Equipment Using Water	3	21	0	7	8
Water Main Challenges/Breaks	1	10	0	8	8
Vibration	3	4	2	9	7

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
