# Peer review of "Water Management for Construction: Evidence for Risk Characterization in Community and Healthcare Settings: A Systematic Review"

_ijerph, 2020, doi:10.3390/ijerph17062168_

Round 1

Reviewer 1 Report

The manuscript is well written and well structured, and the topic is very interesting.

The authors clearly highlight the risk factors related to construction activities that contribute to the growth and spread of waterborne pathogens in both community and healthcare environments. Furthermore, the authors also highlight the aspects that still need to be explored.

The manuscript provides an accurate description of the construction activities associated with waterborne disease cases, so I think it can be extremely useful for workers in the sector such as building owners, facility construction managers, engineers, construction professionals, etc. Finally, I think it can be the starting point for updating guidelines and standards addressing risk management for building water systems.

Author Response

Thank you for your supportive comments for our manuscript.  There were no formal comments to answer.  We appreciate your understanding that this research is positioned to have a positive impact on public health professionals, water management programs, and team members.  

Reviewer 2 Report

A brief summary:

The multidisciplinary Legionella research has developed hand in hand with the level of science and technology, health care and other scientific fields. Monitoring and control technologies are inevitable for the production of safe drinking water. The presented review findings support the need for building owners, water management teams, and public health professionals to address construction activity risk factors and the analysis of current knowledge deficiencies within the scope of an on-going water management program.

Highlighting areas of strength and weakness:

As it is mentioned in limitations: it does not include events included in the press or industry articles, or internet reports from government organizations, or agencies.

Strength: good review of selected articles 

I encouraged you to prepare editorial review bias on articles about Legionella –interesting

Specific comments:

177-179 The demographics associated with waterborne pathogens and construction activities are similar 178 to the demographics for disease cases associated with waterborne pathogens in building water 179 systems. Why you decide to you use this risk management method ? are there other possible ways to evaluate it? (AHP OWA ..etc..) construction activities – it covers a wide range of activities ...can you identify one/ more with the highest risks?

I like the overall study and used methods. I encourage the authors to go through the manuscript and reorganize it a bit (risk management description ..etc). Please avoid repeating statements that are not so important. Check the used methods and clearly describe them.

 Good Luck !

Author Response

Thank you for your supportive comments.  We took your comments and broke the sentences down into 5 key comments we reviewed and made text changes as necessary.  Red text below is a method of highlighting to illustrate text changes. 

COMMENT#1: I encourage you to prepare editorial review bias on articles about Legionella

Response #1

A publication and editorial review bias manuscript about Legionella would be an interesting subsequent article which the authors will take into consideration in the future. In this manuscript the authors supported a comprehensive water management approach encompassing overall water quality. This first requires an awareness and response to a broader range of waterborne pathogens discussed in Gap #4 (Line 411).  By example Table 1 (Line 156) illustrates 84% of the articles focus on Legionella.  Also, Table 1 identifies organisms (Mycobacterium and Sphingomonas) which accounted for significant numbers of disease cases. Furthermore, current water standards and guidelines referenced in the article have an emphasis on Legionella

R#1 Text Changes:  None required.

COMMENT#2: (Line 177-179) The demographics associated with waterborne pathogens and construction activities are similar (Line 178) to the demographics for disease cases associated with waterborne pathogens in building water (Line 179) systems.  

Response #2

We adjusted the text (Line 178) by adding a word for clarification. 

R#2 Text Changes Line 177 - 179: The demographics associated with waterborne pathogens and construction activities are similar to the demographics for disease cases associated with waterborne pathogens in existing building water systems.

COMMENT#3Why was this risk management method used?

Response #3:

To clarify our use of risk management for building water systems we adjusted the key paragraph lines 74 - 89.   To answer you specific question AHP and OWA seem to be safety risk assessment methods more aligned with occupational injury controlled through: a) human judgement factors or b) algorithms for multi-weighted decision making.  These methods and the data to support them were: 1) not available in the articles we reviewed, and 2) not aligned with risk management for building water systems.  By comparison we utilized the risk assessment formula within the context of risk management for building water systems for several reasons:

  1. Alignment with multiple disciplines such as public health, water safety, medicine, infection prevention, epidemiology, and healthcare construction which utilizes risk characterization using variables of frequency and severity.  
  2. Allows correlation between clinical (disease cases and deaths) and environmental (construction activity) data within the accepted water management risk management/risk characterization framework accepted by WHO, CDC, ASHRAE, NSF International, and VHA to evaluate microbial, chemical and physical hazards.
  3. Alignment with risk characterization for human consumption (ingestion and inhalation) of water with significant human health impacts.

R#3 Text Change Line 73 - 85:

These construction activity descriptions, however, lack characterization of the risk. In order for a building owner’s water management team to navigate through a successful risk management process, the team members must understand the frequency of construction activities and clinical severity to determine risk. A similar risk characterization is commonly performed in healthcare settings for airborne pathogens [8]. Risk management for building water systems is a series of systematic steps to reduce risk and is an established process for controlling waterborne pathogens [2,3,7,9].  A risk is the potential to harm humans from exposure (in this context) to a waterborne pathogen [10,11].  Risk is a product of frequency times severity expressed in a formula as:

Risk = frequency x severity

Frequency is a measure of probability. Severity is a measure of consequence.  As frequency and/or severity increase the risk increases. Assessing and managing risk is an essential element of environmental, clinical, and public health practice for implementing a WMP program. 

COMMENT#4Construction activities – it covers a wide range of activities ...can you identify one/ more with the highest risks?

Response #4:

Yes we agree.  This study measured the frequency of occurrence of construction activities associated with waterborne disease cases and deaths to establish severity in order to rank the construction risk factors.  This gives the building owner and construction industry a prioritization of which activity is the highest risk when correlating an environmental factor (construction) with a clinical outcome (disease case or death).  To provide further clarification please see Table 5 on Line 273.  Also this is mentioned both the Discussion section Gap #3 (Line 366) and Conclusions (Line 489) Item #2 (Line 505 - 507).  Text was changed to clarify commissioning activities apply to newly constructed, renovation, unoccupied, etc. building space. 

Text Change Line 505 - 507:  

2) building owners take responsibility to assure a proper building water management commissioning plan is implemented prior to occupancy of any building (e.g. newly constructed, renovated, unoccupied and re-opened, change of use or occupancy, and building acquisition) with an emphasis on healthcare settings for acute patient care services;

COMMENT#5Reorganization regarding risk management

Response #5:

After re-reading the entire manuscript, we have chosen to re-word and adjust paragraphs within lines 73 - 85, rather than reorganize the manuscript.  We believe maintaining the current structure of the article is complementary to the purpose of the study and its title.  We purposefully structured the article about construction activities followed by risk characterization within the overall context of water management.  

R#5 Text Change Line 73 - 85:  See same text change for Comment #3